# Superhard bulk high-entropy carbides with enhanced toughness via metastable in-situ particles

Jiaojiao Hu[1], Qiankun Yang[1], Shuya Zhu[1], Yong Zhang[1], Dingshun Yan[1], Kefu Gan[1] & Zhiming Li [1,2] ✉

Despite the extremely high hardness of recently proposed high-entropy carbides (HECs), the low fracture toughness limits their applications in harsh mechanical environment. Here, we introduce a metastability engineering strategy to achieve superhard HECs with enhanced toughness via in-situ metastable particles. This is realized by developing a (WTaNbZrTi)C HEC showing a solid solution matrix with uniformly dispersed in-situ tetragonal and monoclinic $ZrO_2$ particles. Apart from a high hardness of 21.0 GPa, the HEC can obtain an enhanced fracture toughness of 5.89 MPa·m$^{1/2}$, significantly exceeding the value predicted by rule of mixture and that of other reported HECs. The toughening effect is primarily attributed to the transformation of the metastable tetragonal $ZrO_2$ particles under mechanical loading, which promotes crack tip shielding mechanisms including crack deflection, crack bridging and crack branching. The work demonstrates the concept of using in-situ metastable particles for toughening bulk high-entropy ceramics by taking advantage of their compositional flexibility.

The concept of high-entropy carbides (HECs), proposed based on the original idea of high-entropy alloys (HEAs)[1–4], usually refers to solid solution ceramics containing more than four species of carbides in equimolar or near-equimolar proportions with relatively high configurational entropy[5,6]. In single-phase HECs, metal components randomly occupy the cation sites, and C atoms are located on the anion sublattice. Recently, HECs have attracted significant interests due to their significant properties, e.g., high hardness, low thermal conductivity, excellent thermal stability, outstanding wear resistance, and ultra-high melting point[6–11]. Thus, they are anticipated to be used in diversified extreme conditions, such as in hypervelocity finishing cutter, hypersonic aircraft, nuclear reactor, and refractory crucibles[9,12,13].

To facilitate the design of stable solid solution HECs, several descriptors have been proposed in recent years[14]. For instance, entropy forming ability (EFA) from the first principles has been proposed to address the probability of the formation of single-phase HECs[15]. It has been verified through experiments that compositions with EFA of larger than 50 (eV/atom)$^{-1}$ can form single-phase HECs[16]. Also, other parameters such as valence electron concentration (VEC 8.0 ~ 8.6), average size difference ($\delta_a \leq 5.2\%$), enthalpy of formation ($\Delta H_{mix}$ −15 ~ 5 kJ mol$^{-1}$) have been used for predicting the formation of solid solution HECs[16–21]. Apart from configuration entropy, recent theoretical studies suggest that the vibrational contributions are also important for the phase stability of solid solution HECs[22].

The exceptional properties of the recently designed solid solution HECs distinguish them from the conventional monocarbides. For instance, the (HfTaZrNb)C HEC has an enhanced microhardness of 22.8 GPa at similar deformability compared with the corresponding monocarbides, e.g., TaC (-12.2 GPa), NbC (-14.8 GPa), ZrC (-18.6 GPa) and HfC (-18.3 GPa)[6,23–25]. The (ZrNbTiV)C HEC obtains high microhardness of 20.8 GPa and fracture toughness of 4.7 ± 0.5 MPa·m$^{1/2}$ owing to the solid solution strengthening and in situ nanoplates toughening mechanisms[19]. However, despite the extremely high hardness, most of HECs are predicted to be brittle based on the

[1]Key Laboratory of Nonferrous Metal Materials Science and Engineering (Ministry of Education), School of Materials Science and Engineering, Central South University, Changsha, China. [2]State Key Laboratory of Powder Metallurgy, Central South University, Changsha, China. ✉e-mail: lizhiming@csu.edu.cn

analysis of elastic parameters calculated via the density functional theory (DFT)[9,26]. It has been reported that the brittleness-ductility transition in a (TaNbHfTiZr)C HEC can only occur when the pressure is above 20 GPa due to the strong covalence[18]. In this context, relieving the brittleness and enhancing the toughness of HECs are essentially important, though the research on the toughness optimization of HECs are still quite limited so far.

In this work, we propose to incorporate in situ formed metastable $ZrO_2$ particles into the HECs for significant toughening effect, and this can be achieved by taking Zr as one of the principal elements with the standard powder metallurgy method. We employ in situ metastable $ZrO_2$ particles to toughen HECs, for several considerations: (i) Martensitic transformation of the metastable $ZrO_2$ phase can be triggered during mechanical loading. The transformation process and the associated shielding zones can result in a stress intensity reduction of the crack tip and toughening the bulk material[27–30]; (ii) The mismatches of elastic moduli and thermal expansion coefficients between the $ZrO_2$ particles and the HEC matrix can lead to the presence of residual stress and hence the formation of submicron cracks around the $ZrO_2$ particles upon mechanical loading, providing deflecting and bridging effects on the main cracks[31,32]; (iii) The in situ formation of beneficial $ZrO_2$ particles in the HECs can consume the residual oxygen induced from the powder metallurgy processing, avoiding the formation of the other types of detrimental oxides. We realize the strategy in a promising HEC with nominal composition $(W_{0.2}Ta_{0.2}Nb_{0.2}Zr_{0.2}Ti_{0.2})C$ prepared through ball milling plus spark plasma sintering (SPS). Different sintering pressures were applied to tune the fractions of in situ formed metastable $ZrO_2$ particles and hence the mechanical properties of the bulk HECs.

## Results

### Phase compositions and microstructures

Figure 1 shows the X-ray diffraction patterns of the mixed powders milled for various time periods (i.e., 0, 20, 40, 60, 80, 100, 110 h) and the as-sintered bulk HEC samples. Morphologies of the milled powders are provided in Supplementary Fig. 1. The XRD pattern of the as-mixed raw powders (ball-milling time of 0 h) shows independent diffraction peaks of the five monocarbides (Fig. 1a). As the high-energy ball-milling proceeds, the diffraction peaks of TiC disappear first at 20 h, followed by ZrC at 60 h. An asymmetric peak at the $2\theta$ angle of ~74°, corresponding to TaC and NbC (lattice constants of 4.474 Å and 4.477 Å, respectively)[33], evolves to be symmetric after 20 h ball-milling, indicating the formation of substitutional solid solution. The three main peaks of hexagonal WC are remained even after 110 h of milling, but considerably broadened and weakened due to the potential lattice distortion and formation of nanocrystals. The angular-shaped raw powders turn to be spherical with average size around 200 nm upon ball milling for 110 h (Supplementary Fig. 1). The remaining hexagonal WC phase in the powders suggests the incomplete alloying of the carbides by ball milling.

After sintering at 1800 °C under the pressures of 30 and 50 MPa, a set of peaks for a desired FCC structure are shown in the XRD patterns,

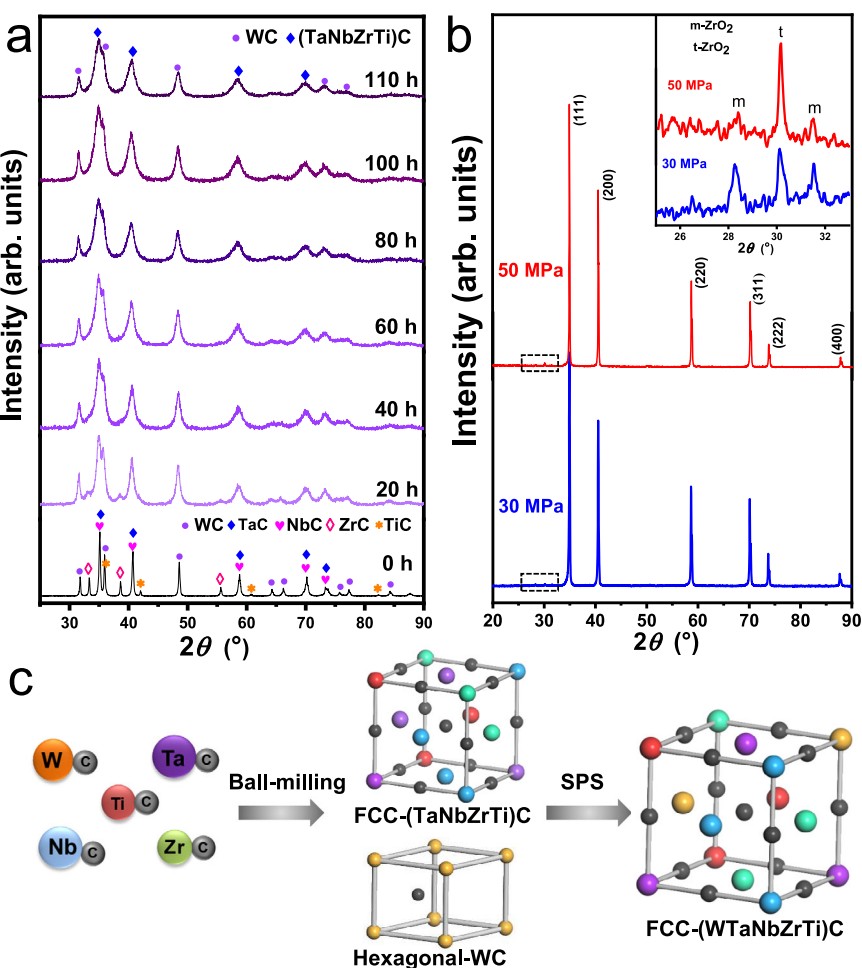

**Fig. 1 | Phase constituents of the mixed powders and sintered bulk samples.** **a** XRD patterns showing the evolution of phase composition in mixed powders with respect to ball-milling time (0, 20, 40, 60, 80, 100, and 110 h). **b** XRD patterns of bulk (WTaNbZrTi)C sintered under different pressures (30 and 50 MPa), the insert shows the enlarged patterns with the $2\theta$ range from 25° to 33°. Source data are provided as a Source Data file. **c** Schematic diagrams showing the phase evolution induced by ball milling and SPS. "m-$ZrO_2$" and "t-$ZrO_2$" in the insert of **b** refer to monoclinic-$ZrO_2$ and tetragonal-$ZrO_2$, respectively.

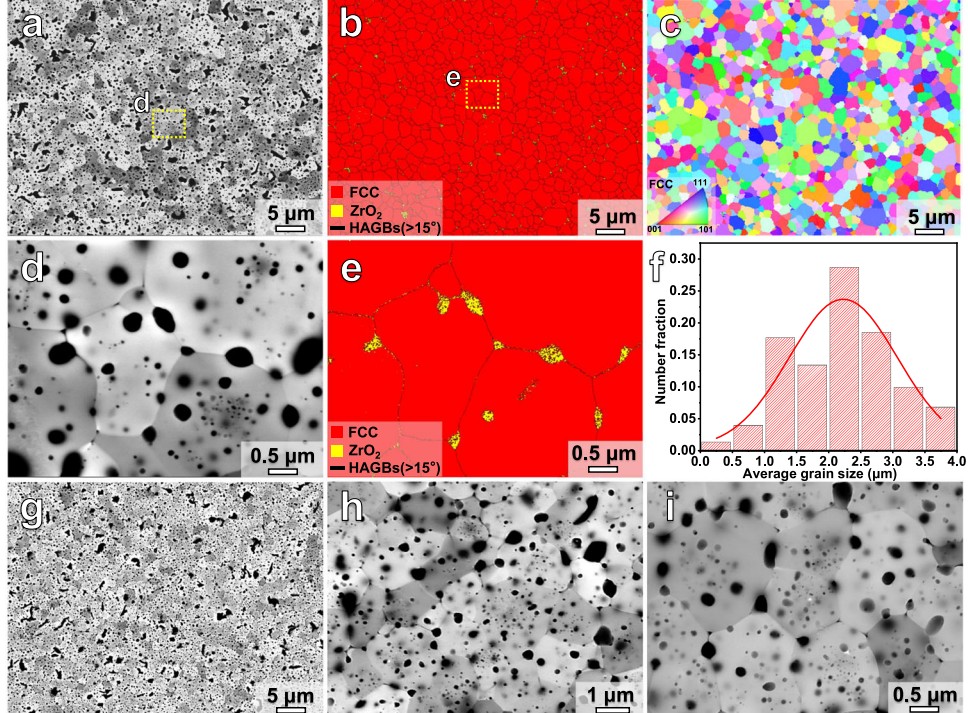

**Fig. 2 | Microstructure of the sintered HEC samples. a, b** BSE image and EBSD phase map of the (WTaNbZrTi)C HEC sintered at 30 MPa; **c** Corresponding EBSD inverse pole figure (IPF) map of the FCC matrix phase; **d** High magnification BSE image of the sample region marked by the dashed yellow rectangle in (**a**); **e** High magnification EBSD phase map of the sample region marked by the dashed yellow rectangle in (**b**), correlated with the sample region in (**d**); **f** Statistical FCC grain size distribution of the HEC sintered at 30 MPa. **g–i** BSE images with different magnifications for the HEC sintered at 50 MPa. "HAGBs" refers to high angle grain boundaries. EBSD analysis can hardly distinguish the tetragonal and monoclinic structures of the ZrO₂ particles, and therefore the specific structures of the ZrO₂ particles are not denoted in the phase maps.

and the peaks of hexagonal WC phase are not presented (Fig. 1b). This suggests that the FCC (WTaNbZrTi)C HEC phase is formed owing to the adequate inter-diffusion between the hexagonal WC and FCC (TaNbZrTi)C during the sintering process. The lattice parameter of the HEC is calculated to be 4.45 Å, close to that of the TaC (4.474 Å ICDD card No. 35-0801). Schematic diagrams illustrating the phase evolution during the powder metallurgy process are shown in Fig. 1c. As schematically illustrated, the mechanical alloying by ball milling leads to the formation of a dual-phase structure consisting of FCC (TaNbZrTi)C and hexagonal WC phases, whereas the high-temperature SPS promotes the final formation of FCC (WTaNbZrTi)C HEC.

Apart from the main FCC (WTaNbZrTi)C HEC phase, the enlarged XRD patterns in the insert of Fig. 1b shows the presence of ZrO₂ with monoclinic (m-) and tetragonal (t-) crystal structures. Based on the intensity variations of the diffraction peaks, the fraction of the metastable t-ZrO₂ phase increases with increasing the sintering pressure from 30 MPa to 50 MPa, and the stable m-ZrO₂ phase becomes almost invisible for the sample sintered at 50 MPa. This suggests that higher sintering pressure is beneficial to the retention of metastable t-ZrO₂ in the HECs. This is consistent with the previous findings about phase transformation of ZrO₂, e.g., the retention of t-ZrO₂ is directly related to the constraint imposed by adjacent matrix grains[34]. The recorded sintering curves are shown in Supplementary Fig. 2. Under the sintering pressure of 50 MPa, the sharper and higher peak of the shrinkage rate curve at the time of around 1200 s suggests faster contraction and higher density of the (WTaNbZrTi)C matrix compared to that under the sintering pressure of 30 MPa. This confirms that the higher sintering pressure promotes the contraction and densification of the HECs, which in turn inhibits the transformation of t-ZrO₂ during subsequent cooling.

Figure 2 displays the BSE and EBSD analysis results for the (WTaNbZrTi)C samples sintered at 30 and 50 MPa. No obvious pores or defects are present in both materials. Nearly ellipsoidal particles displaying a black contrast are dispersed in the HEC matrix, as shown in the BSE images in Fig. 2a, d, g–i. It is worth noting that the particles located in grain boundary regions commonly show larger sizes than that in grain interiors. The corresponding EBSD phase maps confirm the FCC-structure of the HEC matrix, and some relatively larger ZrO₂ particles dispersed in the HEC matrix can be identified (Fig. 2b, e). The inverse pole figure (IPF) map in Fig. 2c suggests that the (WTaNbZrTi)C matrix possesses equiaxed grains with random crystallographic orientation. The size of the FCC grains in the HEC sintered at 30 MPa varies from 0.5 to 4 μm, with an average value of 2.3 μm (Fig. 2f). There is no significant change in the morphology between the HECs sintered at 30 MPa and 50 MPa, but the average grain size of the latter is smaller, i.e., ~1.5 μm. No abnormal growth of FCC matrix grains was observed, which can be partially attributed to the inhibitory effect by the ZrO₂ particles. According to "Image Pro" software analysis of multiple BSE images, the fraction of ZrO₂ particles in the HECs sintered at 30 MPa and 50 MPa can be estimated to be about 6.9% and 6.1%, respectively.

Figure 3 shows the SEM-EDS analysis results of the (WTaNbZrTi)C HEC sintered at 30 MPa. The particles in grain boundary regions are larger than that in grain interiors, and grow toward one side of the matrix grain (Fig. 3a, b). According to the SEM images and EDS line scan results (Fig. 3a–c), a WTaTi-rich phase in pure black contrast is commonly attached to ZrO₂ particle with a distinct interface. The WTaTi-rich phase shows a remarkably smaller size compared to the adjacent ZrO₂ particle. Accordingly, it has a very low fraction and cannot be identified from the XRD patterns in Fig. 1b. The formation mechanism of the WTaTi-rich phase is discussed later. The SEM-EDS mapping shown in Fig. 3d suggests that there is an additional type of rod-shaped phase enriched with C and depleted with Zr, Nb, and Ti. However, the above XRD results only show the FCC HEC matrix and

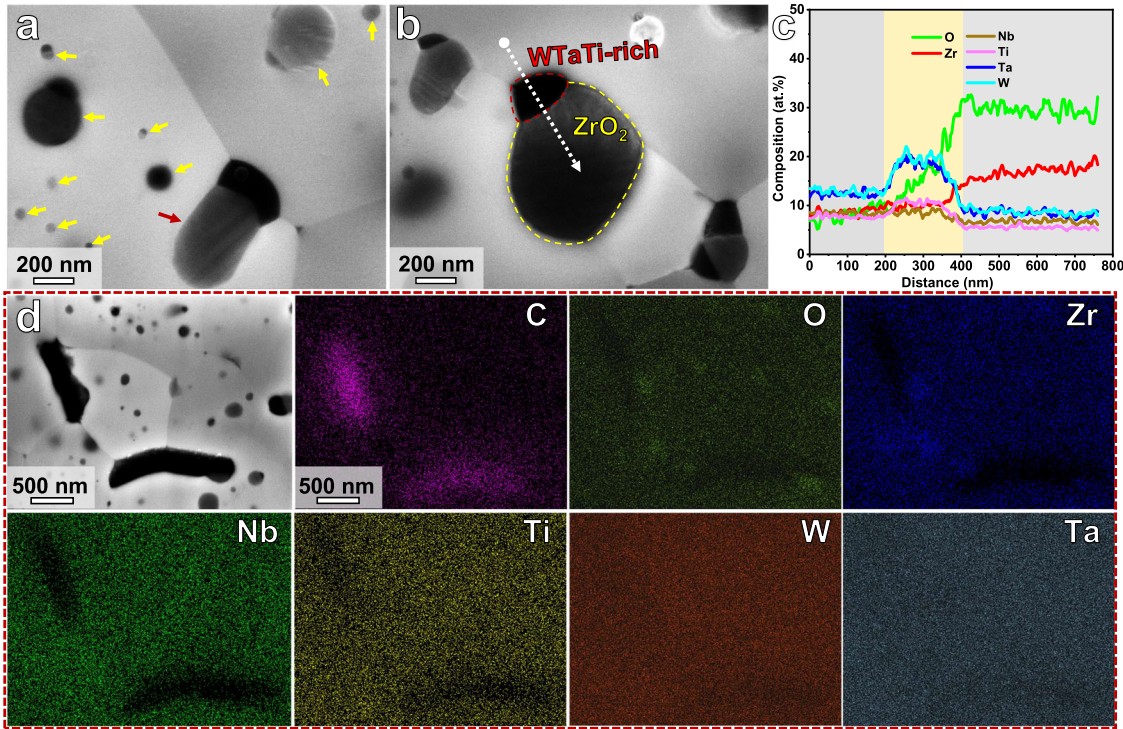

**Fig. 3 | Microstructure of the (WTaNbZrTi)C HEC sintered at 30 MPa. a, b** High magnification SEM images showing the ZrO$_2$ particles in the HEC matrix. **c** Energy dispersive spectrometry (EDS) analysis result of the sample region marked by the dashed white arrow in (**b**). Source data are provided as a Source Data file. **d** Low magnification SEM image, and corresponding EDS mapping. The red and yellow arrows in (**a**) denote the ZrO$_2$ particles located at grain boundary and in grain interiors, respectively.

ZrO$_2$ particles, suggesting that the fractions of the WTaTi-rich and C-rich phases are too low to be detected by these methods.

In order to further reveal the crystal structure and formation mechanism of the ZrO$_2$ particles, TEM/STEM analysis from nano- to atomic-scale has been conducted. Figure 4 shows the typical TEM results for a (WTaNbZrTi)C sample sintered at 30 MPa. The SAED pattern in Fig. 4b collected from the HEC grain and the particles (Fig. 4a), confirms the FCC structure of the HEC matrix and the tetragonal structure of the observed ZrO$_2$ particle. Further, the tetragonal ZrO$_2$ particle is coherent with the HEC matrix. The uniform contrast of the atomic columns in the HAADF-STEM image in Fig. 4c indicates the relatively homogenous distribution of the multiple elements (W, Ta, Nb, Zr and Ti) at nano- and atomic-scales. The interplanar distance of (111) plane in the FCC matrix is 0.263 nm, in agreement with the calculated value of 0.257 nm from the XRD pattern. Lath martensite in the ZrO$_2$ phase can be observed in Fig. 4d. The corresponding SAED pattern in Fig. 4e shows that the observed monoclinic (m-) ZrO$_2$ particle is twinned, and the twin plane is (100). Figure 4f shows the HAADF-STEM image and SAED pattern of another m-ZrO$_2$ particle with twin plane of (011). These observations suggest that, the martensitic transformation of ZrO$_2$ from tetragonal to monoclinic structure is through the twinning-related variant with self-accommodation, which minimizes the shear component of the shape strain[35,36]. The high magnification BF and HAADF images (Fig. 4h, i) show the occupancy of zirconium and oxygen atoms in the monoclinic ZrO$_2$ phase.

Figure 5 presents the STEM-EDS analysis results for a nanosized grain-interior particle. The STEM image in Fig. 5a and the corresponding EDS maps suggest that the nanosized t-ZrO$_2$ phase is coexistent with a WTaTi-rich phase. The coherent interfaces among the HEC matrix, t-ZrO$_2$, and WTaTi-rich phase are distinct and clean (Fig. 5b), indicating a strong interfacial bonding. In addition, at some of the triple junctions of the HEC grains, nanosized W-rich phase is also presented (Supplementary Fig. 3).

## Mechanical properties and fracture behavior

Table 1 lists the mass densities and mechanical properties of the (WTaNbZrTi)C samples with different sintering pressures. The (WTaNbZrTi)C HEC samples sintered at 30 and 50 MPa possess similar mass densities (9.742 and 9.683 g cm$^{-3}$, respectively). However, the hardness, fracture toughness ($K_{IC}$) and compressive strength of the HEC sintered at 50 MPa are notably higher than those of the HEC sintered at 30 MPa. For the hardness values measured under different loads (e.g., 9.8, 49, and 294 N), it is observed that the higher the applied load, the slightly lower the hardness value. The fracture toughness values measured by both single edge notched beam (SENB, ASTEM C 1421-18) and indentation methods are presented in Table 1. The HEC sintered at 50 MPa simultaneously obtains high hardness (HV1) of 21 ± 0.1 GPa and fracture toughness ($K_{IC}$) of 5.89 ± 0.19 MPa·m$^{1/2}$ (via SENB method).

The fracture toughness specimen of the SENB method is shown in Fig. 6a. Figure 6b shows the morphology of a representative Vickers indentation on the polished surface with a load of 49 N. The radius of crack ($c$) satisfying the criteria of $c < 2.5a$ ($a$ is the half diagonal of the Vickers indent), suggesting the suitability of using the Palmqvist crack model in indentation method for evaluating fracture toughness of the HECs. Figure 6c plots the fracture toughness and Vickers hardness values of HECs in the present study and those of the previously reported HECs for a comparative study. The specific values as well as the loads and methods for hardness and fracture toughness measurements of the various HECs and referential monocarbides are listed in Supplementary Table 1. It is worth noting that the fracture toughness value of the current (WTaNbZrTi)C HEC sintered at 50 MPa is up to 73% higher than that estimated from the rule of mixture (ROM), according to the values of the five monocarbides[37–41]. As shown in Fig. 6c, the current HEC is also significantly tougher than the previously reported HECs[19,24,33,42–44]. Such improved toughness of the present HEC is strongly related to the contribution from the in situ formed ZrO$_2$ particles, as discussed later.

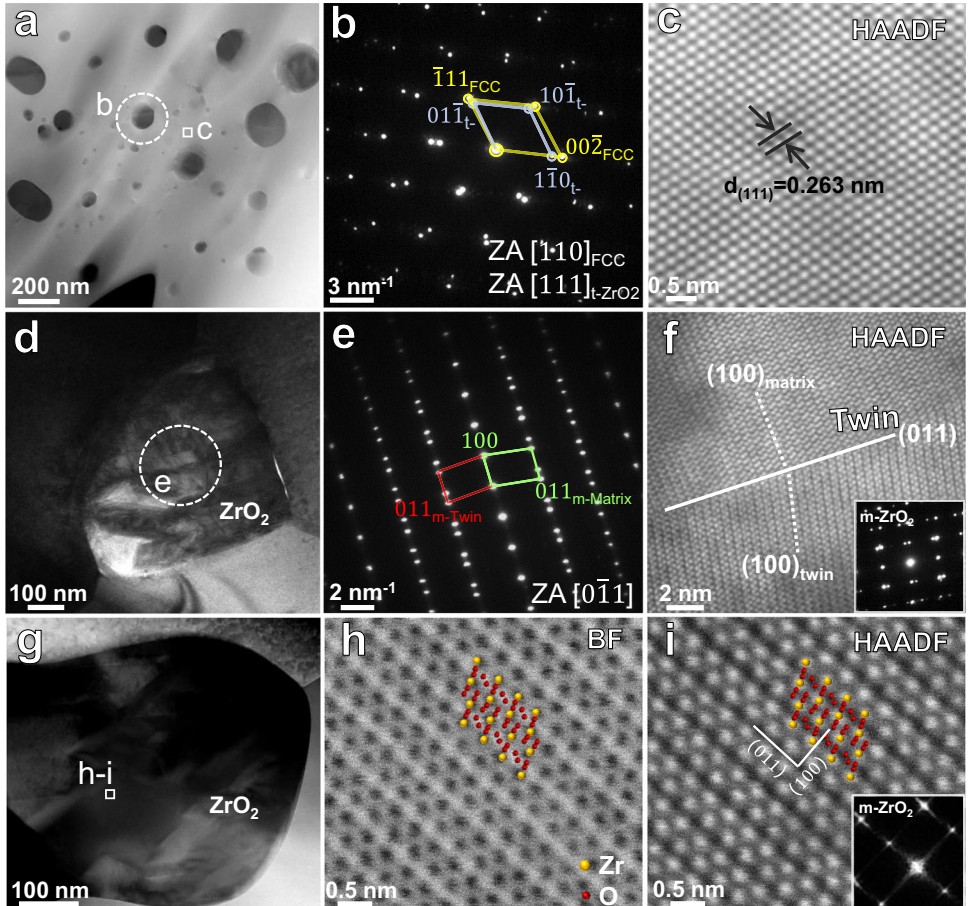

**Fig. 4 | TEM analysis of the (WTaNbZrTi)C HEC sintered at 30 MPa. a** High angle annular dark field (HAADF) scanning TEM (STEM) image. **b** Selected area electron diffraction (SAED) pattern corresponding to the sample region highlighted by the dashed white circle, suggesting the FCC and tetragonal structures of the HEC matrix and ZrO$_2$ particle, respectively. **c** High-resolution HAADF-STEM image of the enlarged area in (**a**), showing the lattice structure of the FCC matrix. **d** Bright field (BF) TEM image of a ZrO$_2$ particle. **e** SAED pattern for the sample region marked by the dashed white circle in (**d**). **f** High-resolution HAADF-STEM image and corresponding SAED pattern showing the twinned structure of a ZrO$_2$ particle. **g** HAADF-STEM image of another ZrO$_2$ particle. **h** High-resolution BF image of the area marked by white rectangle in (**g**). **i** HAADF-STEM image corresponding to (**h**). The lower right insets in (**i**) are corresponding fast Fourier transformation (FFT) image, indicating a monoclinic structure of the presented particle. "m-Matrix" and "m-Twin" in (**e**) refer to the matrix and twin in the monoclinic ZrO$_2$ particle, respectively.

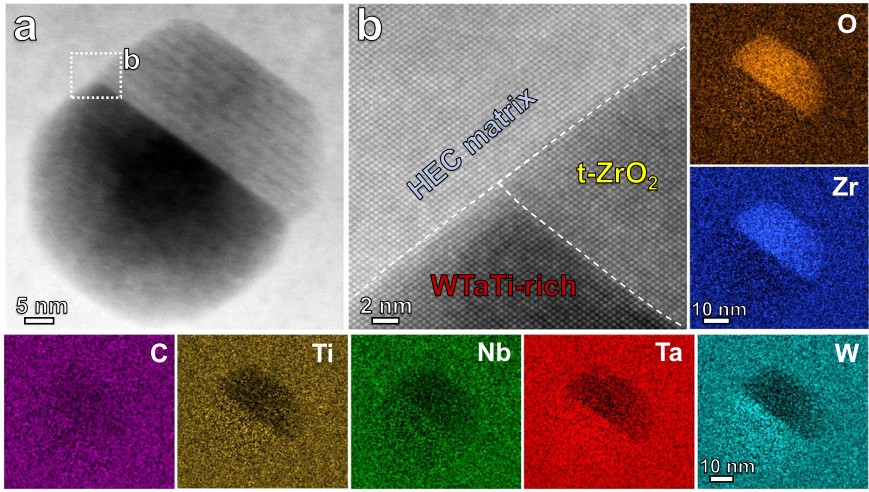

**Fig. 5 | STEM-EDS analysis at near atomic-scale of a nanosized grain-interior particle. a** HAADF-STEM image of a nanosized precipitate region, and the corresponding EDS mappings. **b** Enlarged image of the region marked by white dashed rectangle in (**a**), showing the coherent interfaces among the HEC matrix, t-ZrO$_2$ and WTaTi-rich phases.

The crack propagation behavior of the current HEC is presented by high magnification SEM images in Fig. 6d–g. The radial cracks propagate in zigzag path, and three different pathways for preventing crack propagation in the HEC can be identified from the observations. Firstly, the radial crack is markedly deflected around ZrO$_2$ particles (Fig. 6d), which results in the winding of the propagation path and enhances the crack resistance by consuming more energy. Secondly, crack bridging occurs as demonstrated in Fig. 6e, and the fracture energy is then dissipated through the debonding between ZrO$_2$ particles and HEC matrix as well as the frictional sliding during pullout.

Thirdly, due to the mismatch in thermal expansion coefficients of (WTaNbZrTi)C HEC matrix and ZrO$_2$ phase[45,46], and the twining behavior in the t-ZrO$_2$ phase, microcracks could be formed around some of the ZrO$_2$ particles upon cooling from the sintering temperature[47,48], as demonstrated in Supplementary Fig. 4. These microcracks slowly grow as subcritical cracks under external stress loading, leading to a crack branching effect, as shown in Fig. 6f. This releases part of the strain energy and reduces the stress intensity of the main crack tip, thereby effectively shielding the crack. For the main crack marked in Fig. 6b, it eventually terminates under the interaction with ZrO$_2$ particles, as displayed in Fig. 6g.

Figure 6h, i shows the fracture morphologies of the (WTaNbZrTi)C HEC samples sintered at 30 MPa and 50 MPa, respectively, after compressive testing at room temperature. Both of the HEC samples are characterized by a hybrid of intergranular and transgranular fracture features. Cleavage steps are distributed among different grains with diverse crystallographic orientations, and this phenomenon is more pronounced in the HEC sample with higher sintering pressure (50 MPa). The ZrO$_2$ particles exposed on the fracture surface are mostly intact without rupture, confirming that the ZrO$_2$ phase has remarkably higher resistance to cracking compared to the HEC matrix. This is related to the fact that the deformation energy can be partially absorbed via the stress induced martensitic transformation in the t-ZrO$_2$ phase, as further discussed in the following.

## Discussion

### Formation of the high-entropy solid solution

Significant inter-diffusion between the refractory monocarbides, i.e., TaC, NbC, ZrC, TiC, occurred during mechanical ball-milling to form an FCC solid solution phase, although the hexagonal-WC did not fully

**Table 1 | Mass densities and mechanical properties of the bulk (WTaNbZrTi)C HEC samples sintered under different pressures (30 and 50 MPa)**

| HEC samples | HEC-30MPa | HEC-50MPa |
|---|---|---|
| Mass Density (g cm$^{-3}$) | 9.742 ± 0.083 | 9.683 ± 0.050 |
| Hardness /9.8 N (GPa) | 20.4 ± 0.1 | 21.0 ± 0.1 |
| Hardness /49 N (GPa) | 18.1 ± 0.2 | 18.8 ± 0.1 |
| Hardness /294 N (GPa) | 18.1 ± 0.1 | 18.4 ± 0.1 |
| $K_{IC}$ (MPa·m$^{1/2}$) Shetty Eq. | 5.0 ± 0.2 | 5.4 ± 0.2 |
| $K_{IC}$ (MPa·m$^{1/2}$) Antis Eq. | 7.1 ± 0.5 | 8.2 ± 0.7 |
| $K_{IC}$ (MPa·m$^{1/2}$) SENB method | 5.27 ± 0.25 | 5.89 ± 0.19 |
| Compressive Strength (MPa) | 3323 ± 61 | 3600 ± 95 |

The hardness values measured under different loads (e.g., 9.8, 49, and 294 N), and fracture toughness values tested by different methods are listed.

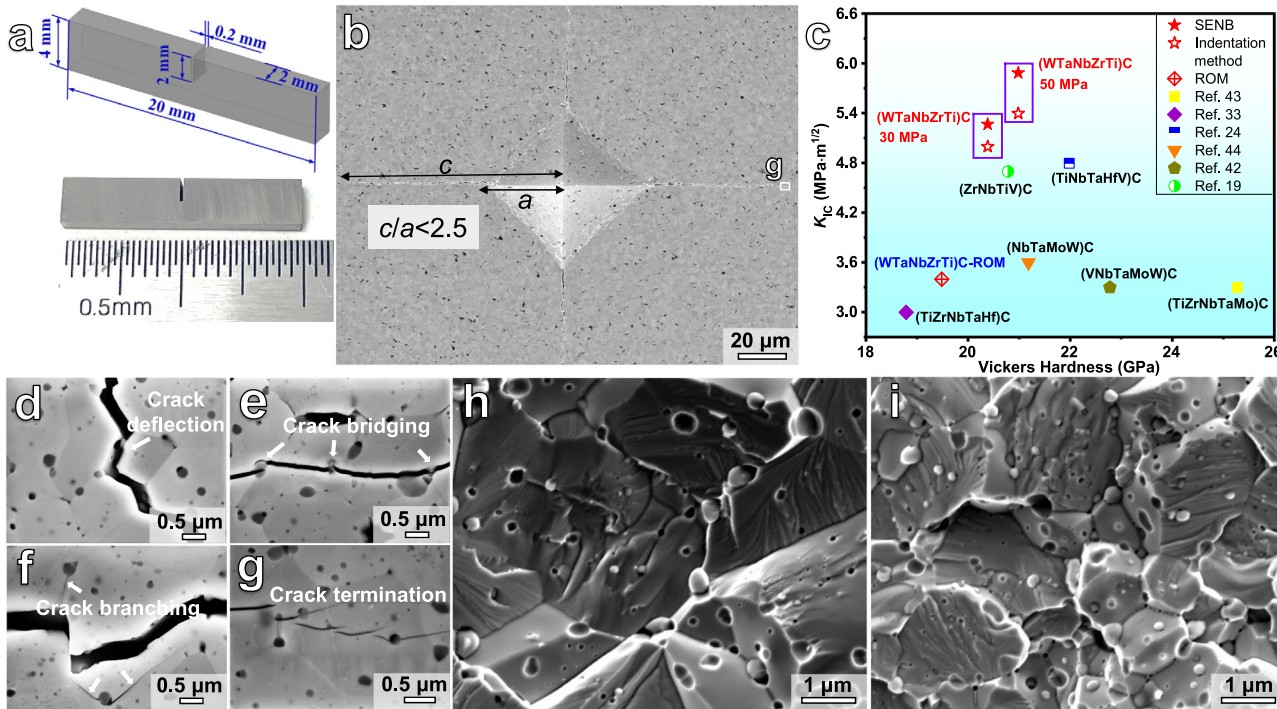

**Fig. 6 | Mechanical properties and crack propagation behavior. a** Sketch and photograph showing the geometry of the specimens for fracture toughness measurement using the single edge notched beam (SENB) method. **b** SEM image of a representative Vickers indentation of the HEC sintered at 50 MPa with a load of 49 N. The annotation "c/a < 2.5" (c is the radius of crack and a is the half diagonal of the Vickers indent) in (**b**) indicates the suitability of using the Palmqvist crack model for evaluating fracture toughness of the present HECs. **c** Fracture toughness versus Vickers hardness values for the HECs in the present study, and the previously reported HECs[19,24,33,42–44], as well as that estimated from the rule of mixture (ROM). The Vickers hardness values plotted in the diagram for the present HECs were measured under a load of 9.8 N. **d**–**f** High magnification SEM images of (WTaNbZrTi)C HEC sintered at 50 MPa, showing the crack propagation behavior. **g** Enlarged image of the region marked by the white rectangle in (**b**), displaying the main crack tip. **h** Fracture morphologies of the HEC sintered at 30 MPa after compressive testing at room temperature. **i** Fracture morphologies of the HEC sintered at 50 MPa.

blend into the FCC phase upon ball-milling. After that, the high-temperature SPS promotes the final formation of the (WTaNbZrTi)C HEC with an FCC solid solution structure. In general, TaC as solvent system can enable larger solubility than that with WC, and it usually tends to act as the host lattice for the formation of the substitutional solid solution[49–51]. The self-diffusion rate of C atoms in transition metal carbides is several orders of magnitude higher than that of the metal components, and is independent of metal's self-diffusion[6]. In the present study, the XRD peaks of TiC, NbC and ZrC disappear sequentially with the increase of ball-milling time (Fig. 1a), indicating an order of Ti>Nb>Zr>Ta for solid state diffusion during mechanical alloying. This is also consistent with the reported phenomena in (HfTaZrTi)C, (HfTaZrNb)C and (TiZrNbTaMo)C HECs during high temperature sintering[6,43].

When it comes to the empirical criteria for predicting the formation of solid solution HECs, several parameters, e.g., average lattice constant difference ($\delta_a$), enthalpy of mixing ($\Delta H_{mix}$), entropy forming ability (EFA) and valence electron concentration (VEC) have been employed in studies of recent years[15,17,19,21]. The average lattice constant difference for HECs can be defined as:[43]

$$\delta_a = \sqrt{\sum_{i=1}^{n} c_i \left[ 1 - \frac{r_i}{\sum_{i=1}^{n} c_i r_i} \right]^2} \qquad (1)$$

where $c_i$ and $r_i$ are the molar concentration and the nearest diffusion distance of metallic atoms in $i$-th monocarbide, respectively. For FCC-TMC (TM = Ta, Nb, Zr, Ti), $r_i = a_i/\sqrt{2}$, where $a_i$ is the lattice constant of the $i$-th monocarbide. For hexagonal WC, $r_i = a_i$[43]. The calculated $\delta_a$ value of the (WTaNbZrTi)C is 4.48%, which fulfills the criterion ($\delta_a \leq 5.2\%$) for forming a single-phase solid solution HEC[21]. The VEC is an enthalpic stability indicator and defined by:[17]

$$VEC = \sum_{i=1}^{n} c_i (VEC)_i \qquad (2)$$

where $c_i$ and $(VEC)_i$ are the molar concentration and VEC of the $i$-th element, respectively. Based on the calculation, the (WTaNbZrTi)C HEC has a VEC value of 8.8. The EFA can be estimated by the energy distribution spectrum of metastable configurations above the zero-temperature ground state[15]. Based on empirical observations, the criterion EFA > 50 (eV/atom)$^{-1}$ for forming a single-phase structure has been recognized[15,16]. Supplementary Table 2 summarizes the VEC, EFA, $\delta_a$ and phase constituents of reported HECs. It has been found that the HECs with EFA < 50 (eV/atom)$^{-1}$, $\delta_a > 5.2\%$ and VEC > 8.8 show multiphase structure. Therefore, in this work, we propose that the three descriptors of EFA ≥ 50 (eV/atom)$^{-1}$, VEC ≤ 8.8, $\delta_a \leq 5.2\%$ can be jointly used to predict the formation of single-phase solid solution structure of HECs. Accordingly, the (W$_{0.2}$Ta$_{0.2}$Nb$_{0.2}$Zr$_{0.2}$Ti$_{0.2}$)C HEC with EFA of 59 (eV/atom)$^{-1}$, VEC of 8.8, and $\delta_a$ of 4.48% has high possibility of forming single-phase solid solution structure. It should be noted that the exact compositions of the FCC solid solution matrix in the present HEC deviate from the nominal values due to the formations of ZrO$_2$ particles and accompanied WTaTi-rich phase with the absorption of oxygen in the fabrication process. The chemical composition of a typical sample region in the FCC HEC matrix measured by STEM-EDS is (W$_{0.20}$Ta$_{0.25}$Nb$_{0.23}$Zr$_{0.13}$Ti$_{0.19}$)C$_{0.94}$ (Supplementary Fig. 5). The VEC of this composition is estimated to be 8.64, and the lattice constant difference is about 4.14% by ignoring the carbon vacancies. This is in line with the above discussed criteria for the formation of solid solution structure.

## Formation and growth mechanisms of the ZrO$_2$ particles

It can be deduced that the oxygen absorbed into the (WTaNbZrTi)C HEC during the fabrication process is mainly trapped by Zr, forming the desired ZrO$_2$ particles, due to the more negative formation energy of ZrO$_2$ (−3.8 eV atom$^{-1}$), compared to that of the other metallic oxides (−3.3 to −2 eV atom$^{-1}$), i.e., WO$_3$, Ta$_2$O$_5$, Nb$_2$O$_5$ and TiO$_2$[52,53]. In the FCC substitutional solid solution of the (WTaNbZrTi)C HEC with TaC as host lattice, Ta and W atoms are more prone to binding for stabilizing the matrix, owing to the more negative mixing enthalpy of Ta-W (−7 KJ mol$^{-1}$), than that of the Ta-Zr, Ta-Nb, Ta-Ti systems with positive mixing enthalpies[54].

To clearly elucidate the growth mechanisms of the ZrO$_2$ particles, a typical sample region containing four ZrO$_2$ particles with sizes ranging from several nanometers to more than hundred nanometers was probed by STEM analysis, as presented in Fig. 7. The four particles of different sizes marked by '1', '2', '3' and '4' in Fig. 7a indicate four different stages of the growth processes, respectively. In an early stage marked by '1', tetragonal ZrO$_2$ particle with a size of about 15 nm is formed beside the WTaTi-rich phase, and a coherent interface in between is present, as shown in Fig. 7b, c. The particle '2' shows the growth of the ZrO$_2$ phase (gray contrast) to a size comparable to that of the adjacent WTaTi-rich phase. As indicated by the STEM-EDS analysis for the particle regions '1', '2', '3', '4' in Fig. 7a-d, the growing rate of the ZrO$_2$ particles is significantly higher than that of the adjacent WTaTi-rich phase. It is worth noting that no other metallic elements are detected in the ZrO$_2$ particles, as confirmed by the EDS line scan results in Fig. 7e. On the one hand, the precipitation of WTaTi-rich phase in the (WTaNbZrTi)C HEC with O impurities (HEC-O) provides interfacial sites for the nucleation and growth of ZrO$_2$ particle. On the other hand, the exsolution of Zr in turn promotes the formation of the WTaTi-rich phase. Accordingly, a symbiotic relationship between the ZrO$_2$ particle and the adjacent WTaTi-rich phase can be identified, as further schematically illustrated in Fig. 7f. The symbiotic mechanism of ZrO$_2$-WTaTi phases demonstrates that the ZrO$_2$ particles are generated in situ during the sintering process, rather than from the oxidation of ZrC powders during the ball milling stage.

It has been established that martensitic transformation of t-ZrO$_2$ to m-ZrO$_2$ takes place when cooling from high temperature to ~1170 °C, accompanied by 3 ~ 5% volume expansion and 0.16 shear strain[55,56]. On cooling from the sintering temperature, the ZrO$_2$ particle expands due to the t → m transformation, producing a hydrostatic pressure within the ZrO$_2$ particle and radial compressive and tangential tensile hoop stresses around the adjacent matrix. In the current HEC with homogeneously dispersed ZrO$_2$ particles, volume dilatation of the ZrO$_2$ particles upon cooling from the sintering temperature can be inhibited by the rigid HEC matrix with high modulus (491.99 GPa)[47], particularly under the high sintering pressure. Hence, the t → m transformation can be partially suppressed, and some t-type ZrO$_2$ particles can be remained at room temperature.

Apart from the constraint of rigid HEC matrix on the t-ZrO$_2$ particles, the refined particle size is also beneficial for the retention of t-ZrO$_2$ at room temperature. The critical crystallize size above which the t → m transformation of ZrO$_2$ would occur, is controlled by the change of total free energy in transformation, including bulk chemical, dilatational, residual shear strain and interfacial energies[47]. Accordingly, the critical crystalline size for pure t-ZrO$_2$ has been reported to be 30 nm[57], while it increases to 600 nm for Al$_2$O$_3$−15 vol% ZrO$_2$ composite[58]. This rationalizes that the larger sized ZrO$_2$ particles at the grain boundaries are more commonly to be transformed into a monoclinic structure, whereas the smaller sized ZrO$_2$ particles (less than ~100 nm, and form a coherent interface with matrix) located in grain interiors usually remain the tetragonal structure, as depicted in Figs. 4 and 5. Further, under a higher sintering pressure of 50 MPa, the faster contraction and higher density of the (WTaNbZrTi)C matrix

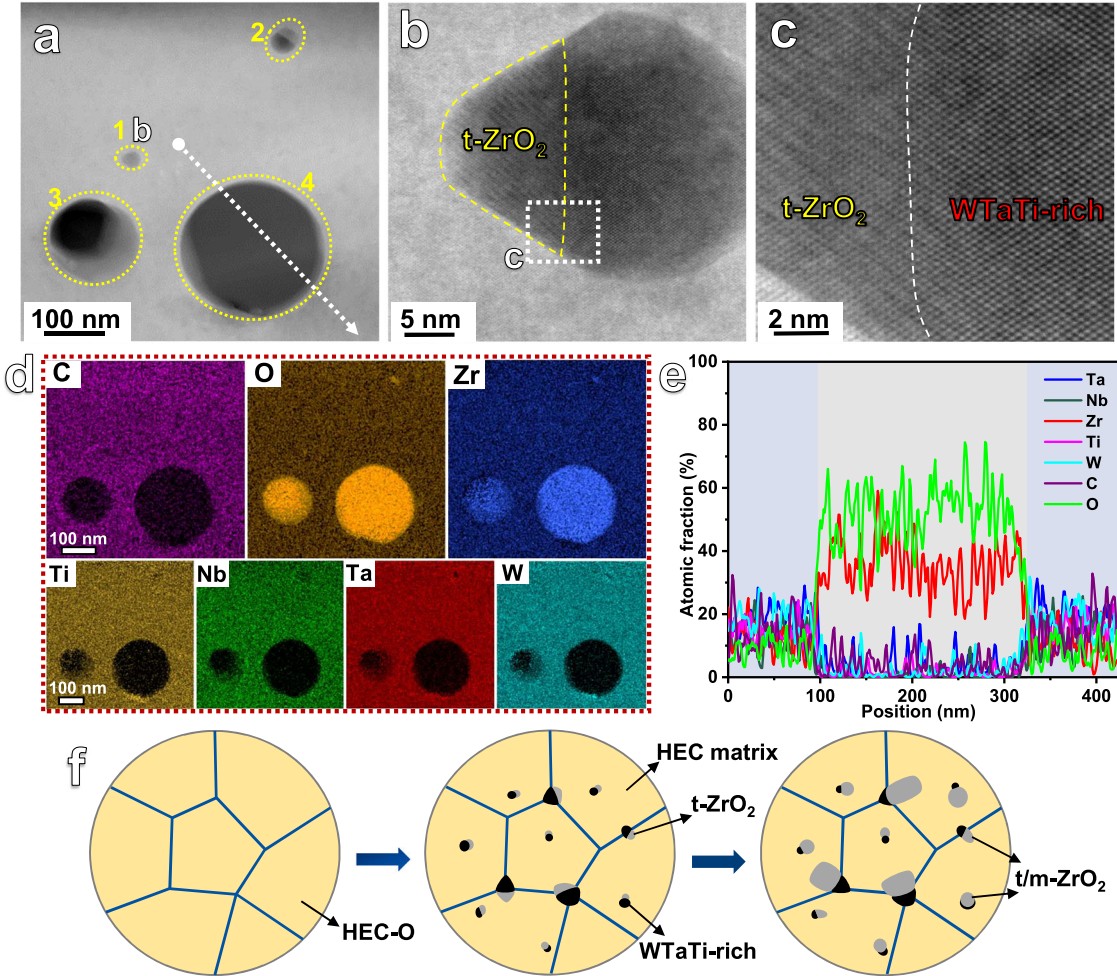

**Fig. 7 | STEM-EDS analysis to elucidate the growth mechanisms of the ZrO₂ particles.** **a** HAADF-STEM image ('1', '2', '3' and '4' refer to four different growth stages of ZrO₂ particles). **b** HAADF-STEM image of a nanosized precipitate region marked by '1' in (**a**). **c** Enlarged image of the region marked by white dashed rectangle in (**b**). **d** EDS mappings of the sample region in (**a**). **e** EDS analysis result of the sample region marked by the dashed white arrow in (**a**). Source data are provided as a Source Data file. **f** Schematic diagrams illustrating the formation and growth processes of the ZrO₂ particles in the present HECs.

contributes to the higher fraction of retained t-ZrO₂ at room temperature during cooling compared to that sintered at 30 MPa.

### Toughening and strengthening mechanisms

To further demonstrate the martensitic transformation of ZrO₂ particles and the associated toughening mechanism in the present (WTaNbZrTi)C HEC, XRD analysis was performed on the fractured surface. XRD patterns of the polished sample surface prior to compressive testing and the fracture surface after compressive testing for the (WTaNbZrTi)C HEC sintered at 50 MPa are shown in Fig. 8a. The t-ZrO₂ has a relative fraction of ~85.3% and is predominant in the as-sintered sample compared to the m-ZrO₂. After fracturing, the relative fraction of t-ZrO₂ decreases to ~66.8% on the fracture surface, implying that about 18.5% of t-ZrO₂ particles in the fracture surface regions have transformed into m-ZrO₂. Accordingly, the (WTaNbZrTi)C HEC sintered at 50 MPa with higher relative fraction of t-ZrO₂ has higher fracture toughness compared to that sintered at 30 MPa. Overall, the higher sintering pressure enhanced toughness of the (WTaNbZrTi)C HEC, arises from the retention of a higher fraction of metastable t-ZrO₂ particles which can absorb energy under mechanical loading by a martensitic transformation to m-ZrO₂ variant.

Figure 8b schematically illustrates the crack propagation in the (WTaNbZrTi)C HEC before final fracture, highlighting the m-ZrO₂ particles (green) transformed from t-ZrO₂ variant (cyan) on the

cracked surfaces. The main crack is deflected with a zigzag path, as also presented in Fig. 6d. In the regions near to the cracked surfaces, t-ZrO₂ particles tend to transform to m-ZrO₂ variant, and the generated residual strain field tends to blunt the main crack.

The contributions of the various mechanisms to the enhanced fracture toughness are also quantitatively evaluated (see Methods). Based on the calculations, the contribution of transformation toughening mechanism is 1.0 MPa·m$^{1/2}$, and the contact shielding of crack bridging accounts for 0.41 MPa·m$^{1/2}$. The microcrack toughening effects including dilatational and modulus contributions are evaluated to be about 0.16 and 0.23 MPa·m$^{1/2}$, respectively. These assessments suggest that the transformation-induced wake zone shielding is most effective for toughening the current HEC among the various mechanisms. In particular, the angular ZrO₂ particles with relatively larger crystalline sizes at the grain boundaries are more susceptible to transformation partially due to the "corner effect"[36]. More specific, facet corners, with pre-existed large residual stress derived from the thermal mismatch between the (WTaNbZrTi)C and the ZrO₂ particles tends to serve as the sites for activating the transformation. Crystalline size effect on the transformation also exists, as discussed above. In addition, the enhanced cracking resistance and fracture energy owing to the presence of metastable tetragonal ZrO₂ particles can prevent the premature fracture of the HEC, leading to the higher hardness and compressive strength (Table 1).

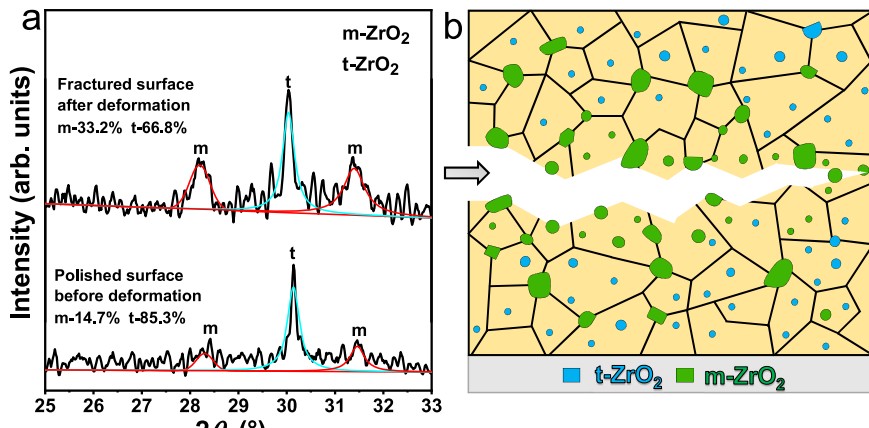

**Fig. 8 | Martensitic transformation of ZrO$_2$ particles induced toughening.**
**a** Fitted XRD patterns of the polished sample surface before compressive testing and the fracture surface after compressive testing for the (WTaNbZrTi)C HEC sintered at 50 MPa. Source data are provided as a Source Data file. **b** Schematic diagram showing the crack propagation and the transformed ZrO$_2$ particles on the cracked surfaces. The gray arrow in **b** points to the direction of crack propagation.

In summary, in situ formed metastable ZrO$_2$ particles toughened bulk (WTaNbZrTi)C high-entropy carbide (HEC) has been successfully developed. In situ formed ZrO$_2$ particles with sizes ranging from several nanometers to submicron are randomly distributed in the HEC matrix. Another WTaTi-rich phase accompanies the ZrO$_2$ particles, and they show a symbiotic relationship. Higher sintering pressure (50 vs. 30 MPa) promotes the retention of a higher fraction of tetragonal ZrO$_2$ particles (compared to the monoclinic variant) to room temperature, due to the higher constrained force from matrix. Apart from a high Vickers hardness of 21.0 GPa (load of 9.8 N), the (WTaNbZrTi)C HEC sintered at 50 MPa exhibits an extraordinary fracture toughness of 5.89 MPa·m$^{1/2}$ (SENB method), exceeding the values calculated from the rule of mixture and that of other reported HECs. This remarkable toughening effect is mainly attributed to the martensitic transformation from the t-ZrO$_2$ to the m-ZrO$_2$ under mechanical loading. Crack deflection, crack bridging and micro-cracking (crack branching) are observed as pathways for inhibiting the main crack propagation, and these are also promoted by the in situ ZrO$_2$ particles. The work thus provides a useful strategy for toughening superhard HECs. For future efforts, the fracture toughness values can be further enhanced based on the present strategy by optimizing the design and processing of the HECs, e.g., adjusting the relative fraction of Zr in the non-equimolar variants, amending the oxygen partial pressure in the sintering chamber and refining the ZrO$_2$ particles in the HEC matrix.

## Methods
### Materials
(W$_{0.2}$Ta$_{0.2}$Nb$_{0.2}$Zr$_{0.2}$Ti$_{0.2}$)C HEC samples were synthesized by a powder metallurgy method. The WC, TaC, NbC, ZrC and TiC powders (Shanghai Shuitian Material Technology Co., Ltd., China) with purity of 99.9% and particle sizes of 1−3 μm were used as the raw materials. The monocarbide powders were mixed in equimolar fraction. Tungsten cemented carbide balls (Yiwu Hongzhou Trading Co., Ltd., China) with various sizes (φ10:φ8:φ5 mm = 1:3:6) were used as milling media in tungsten cemented carbide pots (Changsha Miqi Instrument Equipment Co., Ltd., China) filled with argon. The ball-to-powder weight ratio was set to be 5:1. The dry milling process was conducted using a planetary ball mill (Changsha Miqi Instrument Equipment Co., Ltd., China) at 250 rpm for 110 h. During each cycle of the milling, the ball mill rotates forward for 25 min, pauses for 5 min, and then reverses for 25 min followed by a pause for 5 min. Subsequently, the milled powders were sieved through a 200-mesh sieves, and then put into a graphite die with graphite foil. Spark Plasma Sintering (KCE-FCT HP D 25/4-5D, FCT, Germany) was carried out in

vacuum at 1800 °C with dwell time of 5 min under two different uniaxial pressures, i.e., 30 MPa and 50 MPa. The heating and cooling rate were 85 and 466 °C min$^{-1}$, respectively. Finally, the cylindrical samples with dimensions of φ 30 mm × 8 mm were obtained. Graphite foils on the sintered sample surfaces were removed by grinding with diamond sandpapers.

### Characterization
Phase constituents of the mixed powders and sintered bulk samples were measured by X-ray diffraction (D/max-B 2550, Rigaku, Japan) with a Cu-Kα radiation ($\lambda = 0.154$ nm). The Rietveld refinement method was used to determine the lattice parameters and the phase fractions using MDI Jade 6.0 software. Back-scattered electron imaging (BSEI), electron backscatter diffraction (EBSD), and energy-dispersive X-ray spectroscopy (EDS) analyses on the sintered bulk HECs were carried out using a Tescan Clara scanning electron microscope (SEM). The average grain sizes of the sintered HEC samples were determined based on EBSD measurements using TSL OIM analysis software. High-resolution transmission electron microscopy (TEM) and scanning TEM (STEM) analyses were conducted on lamellae specimens prepared by ion milling, using a FEI Talos F200x microscope operating at 200 KV. Nano-scale elemental distributions in the HEC samples were also examined by STEM-EDS analysis. The mass density of cleaned bulk HEC specimens was measured by Archimedes' method.

Hardness measurements were performed on the Vickers micro-hardness tester (FUTURE, FM-ARS9000, Japan) with a dwell time of 10 s at 9.8 N, 49 N and 249 N, respectively. At least five indentations were performed for each condition to obtain the standard errors of the measurements. The Vickers indentation with the load of 49 N was also used to estimate the fracture toughness, according to the following Shetty Eq. (3) for Palmqvist cracks[59], and Antis Eq. (4) for median cracks[19], respectively:

$$K_{\mathrm{IC}} = 0.0028 \sqrt{\frac{HP}{\sum_{i=1}^{4} l_i}} \quad (3)$$

$$K_{\mathrm{IC}} = 0.016 \left(\frac{E}{H}\right)^{1/2} \frac{P}{c^{3/2}} \quad (4)$$

where $H$ is the Vickers hardness (N mm$^{-2}$), $P$ is the applied load of 49 N and $l$ is the radical crack length starting from the indentation corners

(mm). $c$ in Eq. (4) is the average crack length ($c = 0.25 \sum_{i=1}^{4} l_i$), $E$ is the Young's modulus. Young's modulus ($E$) and Poisson's ratio ($v$) were determined using an ultrasonic method based on the longitudinal and shear wave velocities (ASTM C 1419-14). Furthermore, the limited wake zones of the martensitic transformation surrounding the cracks was detected using the atomic force microscopy (AFM) method (Rigaku, AFM5300E). For a comparative study, the fracture toughness was also determined using a universal testing machine (23 MTS Insight) based on the single edge notched beam (SENB, ASTM C 1421-18) method. The dimensions of the SENB specimens are $2\,mm \times 4\,mm \times 20\,mm$ (width × height × length), with a notch of $2\,mm \times 0.2\,mm$. The $K_{IC}$ was calculated by Eqs. (5) and (6):[60]

$$K_{IC} = \frac{PS}{BW^{\frac{3}{2}}} f_1\left(\frac{a}{w}\right) \tag{5}$$

$$f_1\left(\frac{a}{w}\right) = 2.9\left(\frac{a}{W}\right)^{\frac{1}{2}} - 4.6\left(\frac{a}{w}\right)^{\frac{3}{2}} + 21.8\left(\frac{a}{w}\right)^{\frac{5}{2}} - 37.6\left(\frac{a}{w}\right)^{\frac{7}{2}} + 38.7\left(\frac{a}{w}\right)^{\frac{9}{2}} \tag{6}$$

where $P$ is the applied load, $S$ is the span length, $a$ is the notch depth, $B$ and $W$ are the thickness and the width of specimens, respectively. To prepare cylinder samples for compression test, the bulk HEC samples were wire cut and machined into a dimension of $\varphi\,6\,mm \times 6\,mm$. Compression tests were conducted using an Instron 3369 instrument at a strain rate of $10^{-3}\,s^{-1}$ at room temperature.

## Calculation of toughening mechanisms

The fracture toughness ($K_{IC}$) of the current HEC is quantitatively evaluated by considering the contributions of crack tip shielding mechanisms, such as transformation toughening, microcrack toughening, and ductile particle bridging. For a steady-state crack with a constant wake zones width, the increased toughness ($\Delta K_T$) due to the transforming t-$ZrO_2$ particles can be written by:[36,61]

$$\Delta K_T = \frac{\eta V_f E \epsilon^T \sqrt{h}}{(1 - \nu)} \tag{7}$$

where the constant $\eta$ is dependent upon the shape of the zone ahead of the crack tip. $E$ is Young's modulus. $v$ is Poisson's ratio of the composite material. $\varepsilon^T$ and $h$ are the isotropic dilatational strain and width of the process zone, respectively. $V_f$ is the volume fraction of the transformed t-$ZrO_2$ at the fracture surface. Accordingly, the evaluation is conducted by taking $\eta = 0.38$, $V_f \approx 18.5\%$, $E = 491.99$ GPa, $v = 0.174$, $\varepsilon^T = 0.05$[62].

The transformation zone about the Vickers indentation can be measured directly using the AFM method[63], as shown in Supplementary Fig. 6. The critical transformation stress is given by[64]

$$\sigma_c^T \cong H(d/a)^3 \tag{8}$$

$$h = \frac{2}{9\pi} \left[\frac{3K_m(1+\nu)}{\sigma_c^T}\right]^2 \tag{9}$$

where $H$ is the indentation hardness (HV30 ∼ 18.4 GPa), $d$ is the indentation diagonal length (∼173 μm), $a$ is the transformation zone diameter (∼243 μm), $K_m$ is the matrix toughness, taken as 3.4 MPa·m$^{1/2}$ (the ROM value). The width of the process zone ($h$) is about 0.23 μm. Thus, $\Delta K_T$ is estimated to be about 1.00 MPa·m$^{1/2}$.

As a result of thermal expansion mismatch (anisotropy) and/or transformation strain during the sintering cooling, the preexisting microcracks can generate a larger wake zone, shielding the local crack propagation and achieving toughening[65]. There are two sources of the microcrack toughening mechanism. One is the reduced effective elastic moduli. The other is the dilatation when micro-cracking, and

the corresponding extent of crack shielding is given as:[32]

$$\Delta K_{c1} = 0.32 E \theta_T \sqrt{h} \tag{10}$$

$$\theta_T = \frac{16(1 - \nu^2)\eta\sigma}{3E} \tag{11}$$

$$\sigma = \frac{2E\varepsilon^T}{9(1 - \nu)} \tag{12}$$

where $\eta$ is the number density of the microcracked particles, $\sigma$ is the residual tension that induces microcracks. Contribution of reduced modulus to toughness can be calculated by $\Delta K_{c2} = 1.42\eta K_c$[66]. Based on the observation for crack propagation, the number density of the microcracked $ZrO_2$ particle is estimated to be 0.03, hence the $\Delta K_{c1}$ and $\Delta K_{c2}$ are 0.16 and 0.23 MPa·m$^{1/2}$, respectively.

It should be noted that when the critical stress for microcracking is lower than that for transformation, microcracking accompanies the transformation under mechanical loading. The combination of the two shielding mechanisms gives the greatest toughening. In this case, the increment of toughness ($\Delta K$) cannot be separated into the two independent contributions, but the interaction stress should be considered[36].

The contribution of the crack bridging process $\Delta K_{cb}$ can be calculated by:[67]

$$\Delta K_{cb} = 2.5 V E \Delta\alpha\Delta T \sqrt{R} \tag{13}$$

where $V$ is the volume fraction of bridging particles of radius $R$, $\Delta\alpha$ is the thermal expansion mismatch (-6.99 × 10$^{-6}$ K$^{-1}$) and $\Delta T$ is the temperature change during cooling from which the residual stress is developed. Based on the calculation, the $ZrO_2$ particle bridging contributes to the enhancement of fracture toughness as about 0.41 MPa·m$^{1/2}$.

## Data availability

All data needed to evaluate the conclusions are present in the paper and the Supplementary Materials. Source data are provided with this paper.

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

## Acknowledgements

The financial support by the Natural Science Foundation of Hunan province in China (Grant No. 2021JJ10056), the Science and Technology Innovation Program of Hunan Province (Grant No. 2023RC1013) and the National Natural Science Foundation of China (Grant No. 51971248) are gratefully acknowledged.

## Author contributions

Z.L. conceived the idea and supervised the project. J.H. prepared the materials, performed the mechanical tests, and conducted the XRD, SEM/EBSD characterization. Q.Y. and S.Z. conducted the TEM/STEM. Y.Z., D.Y., and K.G. contributed to the data analysis. J.H., Y.Z., and Z.L. wrote the manuscript. All authors contributed to the discussion of the results and commented on the manuscript.

## Competing interests

The authors declare no competing interests.
