## [Peer Review File · Nature Communications]

Superhard bulk high-entropy carbides with enhanced toughness via metastable in-situ particlesREVIEWER COMMENTS

Reviewer #1 (Remarks to the Author):

Comment 1: In general, the language of this manuscript needs to be improved.

Comment 2: Page 2 line 36, please correct the spelling of “theoretical”.

Comment 3: Please revise the sentence “For instance, the (HfTaZrNb)C ... corresponding monocarbides”. Give some microhardness values of the monocarbides.

Comment 4: If possible, please add the vendors of the carbide raw materials, tungsten cemented carbide media and pots.

Comment 5: Was the starting powders dry milled? During the planetary ball milling, was the rotation direction reversed after each cycle? How was the average grain size calculated based on the inverse pole figure? Please mention.

Comment 6: For the HEC sintering, what was the heating and cooling rate? After SPS, how did the author remove the graphite foil? Please add.

Comment 7: How were the lattice parameter and the quantity of t/m-ZrO₂ calculated? Rietveld refinement based on XRD patterns? Please add in the Characterization section.

Comment 8: Page 14 line 396, c is in Equation (2); line 404, KIC should be calculated by Equation (5) and (6).

Comment 9: Revise the caption of Fig. 3. a-b is the SEM images of the what?

Comment 10: Page 6 line 149, (00-2)FCC//(1-10)t-ZrO₂ was not given in Fig 4, remove it.

Comment 11: Page 12 line 322, please revise the sentence “The (WTaNbZrTi)C HEC ... 30 MPa”. The comma should be removed.

Reviewer #2 (Remarks to the Author):

Interesting original manuscript which first time describing the concept of transformation toughening in bulk high-entropy ceramics by in - situ ZrO₂ grains!

The work support the conclusions, the methodology is OK with enough details with excellent HRTEM work.

I have several comments:

1. The title - Ultrahard bulk high-entropy carbides with enhanced toughness via metastable in-situ particles.....

According to me the hardness measured at 9.8 N with value 21 GPa is not ultrahard.... please see publications:

J. Dusza, T. Csanádi, D. Medved', R. Sedlák, M. Vojtko, M. Ivor, H. Ünsal, P. Tatarko, M. Tatarková, P. Šajgalík, Nanoindentation and tribology of a (Hf-Ta-Zr-Nb-Ti)C high-entropy carbide, J. Eur. Ceram. Soc.

41 (2021) 5417–5426A.

Naughton Duszová, L. Ďáková, T. Csanádi, A. Kovalčíková, V. Kombamuthu, H. Ünsal, P. Tatarko, M. Tatarková, P. Hvizdoš, P. Šajgalík, Nanohardness and indentation fracture resistance of dual-phase high-entropy ceramic, *Ceram. Int.*, on line, <https://doi.org/10.1016/j.ceramint.2022.12.027>

reporting hardness of HE ceramics measured at 9.8 N with significant higher hardness.

2. In the methods - For a comparative study, the fracture toughness was also determined using an 402 universal testing machine (23 MTS Insight) based on the single edge notched beam (SENB, 403 ASTM C 1421-18) method. The dimension of testing samples is 2 mm × 4 mm × 20 mm, with 404 a pre-fabricated crack of 2 mm × 0.2 mm.....

In this case it is not "crack" but notch, with a radius cca 0.1 mm, which is a little large value which can cause increased measured fracture toughness value.....

3. what is Your idea, how it is possible to optimize the ZrO₂ content with the aim to optimize the fracture toughness value?

Response to Reviewers' Comments

NCOMMS-23-09693

We would like to begin by thanking the editor and reviewers for the valuable suggestions and comments. Our response is structured as follows: The comments from the reviewers are copied below (black, italic font). For each comment, we present a response and the corresponding manuscript modifications (blue font). The amended manuscript is enclosed. The changes therein are shown in red font.

Reviewer #1

Comment 1: In general, the language of this manuscript needs to be improved.

Response: We thank the reviewer for carefully reading our paper and for making helpful comments. We have made efforts to improve the language of the manuscript, and some main modifications are listed below.

Modifications:

1. We have revised the title to: “Superhard bulk high-entropy carbides with enhanced toughness via metastable in-situ particles”. On page 1, we revised the associated sentences in the abstract: “Here, we introduce a strategy to achieve superhard HECs with enhanced toughness via in-situ metastable particles”, and “The work demonstrates the concept of using in-situ metastable particles for toughening bulk high-entropy ceramics by taking advantage of their compositional flexibility”.
2. In line 36 of page 2, “theretical” has been revised to “theoretical”.
3. On page 3, the original sentence “The process and shielding zones results in a stress intensity reduction of the crack tip and toughening the bulk material” has been revised to “The transformation process and the associated shielding zones can result in a stress intensity reduction of the crack tip and toughening the bulk material”. The original sentence “The mismatch of elastic modulus and thermal expansion coefficient between the ZrO₂ particles and the HEC matrix leads to the presence of residual stress and the formation of submicron cracks around the ZrO₂ particles upon mechanical loading” has been revised to “The mismatches of elastic moduli and thermal expansion coefficients between the ZrO₂ particles and the HEC matrix can lead to the presence of residual stress and hence the formation of submicron cracks

- around the ZrO₂ particles upon mechanical loading”. On page 3, the expression “the residual” has been added in the sentence “The in-situ formation of beneficial ZrO₂ particles in the HECs can consume the residual oxygen induced from the powder metallurgy processing, avoiding the formation of the other types of detrimental oxides”.
4. On page 4, according to the formatting instructions, the expression “supplementary Fig. S1” has been revised to “Supplementary Fig. 1”. Additionally, the original sentence “Morphologies of the powders are provided in supplementary Fig. S1” has been revised to “Morphologies of the milled powders are provided in Supplementary Fig. 1”. On page 4, the original sentence “The three main peaks of hexagonal WC are still maintained even after 110 h of milling” has been revised to “The three main peaks of hexagonal WC are remained even after 110 h of milling”.
 5. On page 5, the expression “at different pressures” in the original sentence “The recorded sintering curves at different pressures are shown in Supplementary Fig. 2” has been removed.
 6. On page 6, the expression “with an orientation relationship of $(00\bar{2})_{\text{FCC}}//(\bar{1}\bar{1}0)_{\text{t-ZrO}_2}$ and $[110]_{\text{FCC}}//[111]_{\text{t-ZrO}_2}$ ” has been removed.
 7. On page 7, the comma in the original sentence “The fracture toughness values measured by both single edge notched beam (SENB, ASTM C 1421-18) and indentation methods, are presented in Table 1” has been removed.
 8. On page 8, the original sentence “Fig. 6h and 6i shows the fracture morphologies of the (WTaNbZrTi)C HEC samples sintered at 30 MPa and 50 MPa, respectively” has been revised to “Fig. 6h and 6i show the fracture morphologies of the (WTaNbZrTi)C HEC samples sintered at 30 MPa and 50 MPa, respectively”. On page 8, the original sentence “A typical brittle fracture behavior characterized by a hybrid of intergranular and transgranular fracture features is evident for both the two HEC samples” has been revised to “Both of the HEC samples are characterized by a hybrid of intergranular and transgranular fracture features.” On page 8 (Discussion part), “Formation of the high-entropy ceramic solid solution” has been revised to “Formation of the high-entropy solid solution”.
 9. On page 9, the expression “component” in the original sentence “and the nearest diffusion distance of metallic atoms in *i*-th component monocarbide” has been removed. The expression “For hexagonal WC, $r = a$ ” has been revised to “For hexagonal WC, $r_i = a_i$ ”.
 10. On page 10, the expression “the three descriptors of $\text{EFA} > 50 \text{ (eV/atom)}^{-1}$, $\text{VEC} < 8.8$, $\delta_a < 5.2\%$ ” has been revised to “the three descriptors of $\text{EFA} \geq 50 \text{ (eV atom}^{-1}\text{)}^{-1}$, $\text{VEC} \leq 8.8$, $\delta_a \leq$

5.2%”. The original sentence “Accordingly, the $(W_{0.2}Ta_{0.2}Nb_{0.2}Zr_{0.2}Ti_{0.2})C$ HEC with EFA ~ 59 (eV/atom) $^{-1}$, VEC ~ 8.8 , $\delta_a \sim 4.48\%$ has high possibility of forming single-phase solid solution structure” has been revised to “Accordingly, the $(W_{0.2}Ta_{0.2}Nb_{0.2}Zr_{0.2}Ti_{0.2})C$ HEC with EFA of 59 (eV/atom) $^{-1}$, VEC of 8.8 and δ_a of 4.48% has high possibility of forming single-phase solid solution structure”. The comma in the original sentence “It should be noted that, the exact compositions of the FCC solid solution matrix” has been removed.

11. On page 11 of the revised manuscript, the original sentence “the precipitation of WTaTi-rich phase region in the $(WTaNbZrTi)C$ HEC with O impurities (HEC-O), provides interfacial sites for the nucleation and growth of ZrO_2 particle” has been revised to “the precipitation of WTaTi-rich phase in the $(WTaNbZrTi)C$ HEC with O impurities (HEC-O) provides interfacial sites for the nucleation and growth of ZrO_2 particle”. On page 11, the original sentence “the exsolution of Zr in turn promotes the formation of the WTaTi-rich phase, enabling their symbiotic relationship as schematically illustrated in Fig. 7f” has been revised to “the exsolution of Zr in turn promotes the formation of the WTaTi-rich phase. Accordingly, a symbiotic relationship between the ZrO_2 particle and the adjacent WTaTi-rich phase can be identified, as further schematically illustrated in Fig. 7f”.
12. On page 11, the expression “It is generally acknowledged that martensitic transformation” has been revised to “It has been established that martensitic transformation”. The original sentence “some t-type ZrO_2 particles can be maintained at room temperature” has been revised to “some t-type ZrO_2 particles can be remained at room temperature”.
13. On page 12, the original sentence “In order to confirm the martensitic transformation of ZrO_2 particles induced toughening mechanism” has been revised to “To further demonstrate the martensitic transformation of ZrO_2 particles and the associated toughening mechanism”. On page 12, the original sentence “The $(WTaNbZrTi)C$ HEC sintered at 50 MPa with higher relative fraction of t- ZrO_2 , has higher fracture toughness compared to that sintered at 30 MPa” has been revised to “Accordingly, the $(WTaNbZrTi)C$ HEC sintered at 50 MPa with higher relative fraction of t- ZrO_2 has higher fracture toughness compared to that sintered at 30 MPa”.
14. On page 13 of the revised manuscript, the original sentence “The enhanced cracking resistance and fracture energy owing to the presence of metastable tetragonal ZrO_2 particles can also prevent the premature fracture of the HEC” has been revised to “Additionally, the enhanced cracking resistance and fracture energy owing to the presence of metastable tetragonal ZrO_2 particles can prevent the premature fracture of the HEC”. On page 13 (Summary part), the

original sentence “in-situ formed metastable ZrO₂ particles toughened bulk (WTaNbZrTi)C high-entropy carbide (HEC) was successfully developed” has been revised to “in-situ formed metastable ZrO₂ particles toughened bulk (WTaNbZrTi)C high-entropy carbide (HEC) has been successfully developed”. The original sentence “Apart from a high hardness of 21.0 GPa, the (WTaNbZrTi)C HEC sintered at 50 MPa exhibits an extraordinary fracture toughness of 5.89 MPa·m^{1/2},” has been revised to “Apart from a high Vickers hardness of 21.0 GPa (load of 9.8 N), the (WTaNbZrTi)C HEC sintered at 50 MPa exhibits an extraordinary fracture toughness of 5.89 MPa·m^{1/2} (SENB method)”.

15. On page 14, the expression “*c* in equation (2)” has been revised to “*c* in equation (4)”. On page 15, the original sentence “The *K_{IC}* was calculated by Equation (3) and Equation (4)” has been revised to “The *K_{IC}* was calculated by Equation (5) and Equation (6)”.
16. On page 16, the original sentence “the preexisting microcracks are capable of generating a larger wake zone” has been revised to “the preexisting microcracks can generate a larger wake zone”.
17. On page 17, the expression “roughly” in the original sentence “the number density of the microcracked ZrO₂ particle is roughly estimated to be 0.03, hence the ΔK_{c1} and ΔK_{c2} are 0.16 and 0.23 MPa·m^{1/2}, respectively” has been removed. The original sentence “It should be noted that when the critical stress for microcracking is lower than that of transformation” has been revised to “It should be noted that when the critical stress for microcracking is lower than that for transformation”.
18. On page 23, the figure caption “the insert shows the enlarged patterns from the 2θ range from 25° to 33°” has been revised to “the insert shows the enlarged patterns with the 2θ range from 25° to 33°”.
19. On pages 23~30, according to the formatting instructions, the original figure captions “Fig. 1, Fig. 2.....Fig. 8” have been revised to “Figure 1, Figure 2.....Figure 8” in the revised manuscript.
20. On page 24, the original figure caption “Fig. 3. Microstructure of the (WTaNbZrTi)C HEC sintered at 30 MPa. a-b, SEM image of the” has been replaced by the “Figure 3. Microstructure and element distribution in the (WTaNbZrTi)C HEC sintered at 30 MPa. a-b, High magnification SEM images showing the ZrO₂ particles in the HEC matrix”.

21. On page 30, the original figure caption “The martensitic transformation of ZrO₂ particles induced toughening (WTaNbZrTi)C HEC” has been revised to “Martensitic transformation of ZrO₂ particles induced toughening”.
22. On page 31, the original sentence “Mass densities and mechanical properties of the bulk (WTaNbZrTi)C HEC samples sintered under different pressures (30 and 50 MPa)” has been bolded as a short title of Table 1.
23. The serial numbers of the original references have been changed in the revised manuscript and supplementary information due to the adding of new references (Ref. 10, Ref. 11, and Ref. 25).
24. In supplementary information, on page 8, the sentence: “The Vickers hardness and fracture toughness of individual monocarbides and various HECs” has been bolded as a short title of the supplementary Table 1.
25. In supplementary information, on page 9, the sentence: “Structural parameters of the various reported HECs” has been added as a short title of the supplementary Table 2. In addition, “atomic-size difference (δa)” has been revised to “atomic-size difference (δ_a)”. The expression “Single or multiple Phase” has been revised to “Single or Multiple Phase” in the revised supplementary information.
26. According to the journal’s formatting instructions “the express inverse unit dimensions using negative integers”, the expressions “KJ/mol” and “g/cm³” have been revised to “KJ mol⁻¹” and “g cm⁻³”, respectively, in the revised manuscript and supplementary information.

Comment 2: Page 2 line 36, please correct the spelling of “theoretical”.

Response: We appreciate the reviewer’s suggestion, and apologize for the typo. The spelling of “theoretical” has been corrected as suggested by the reviewer.

Modification: In line 36 of page 2, “theretical” has been revised to “theoretical”.

Comment 3: Please revise the sentence “For instance, the (HfTaZrNb)C ... corresponding monocarbides”. Give some microhardness values of the monocarbides.

Response: Thanks for the kind suggestion. The microhardness values of the monocarbides reported in Ref. 24 and Ref. 25 have been added in the revised sentence.

Modification: On page 2, the original sentence “For instance, the (HfTaZrNb)C HEC has an enhanced microhardness of 22.8 GPa at similar deformability compared with the corresponding monocarbides” has been revised to “For instance, the (HfTaZrNb)C HEC has an enhanced microhardness of 22.8 GPa at similar deformability compared with the corresponding monocarbides, e.g., TaC (~12.2 GPa), NbC (~14.8 GPa), ZrC (~18.6 GPa) and HfC (~18.3 GPa)”. In addition, the reference (D. Sciti, et al., *J. Am. Ceram. Soc.* 2008, 91: 1433-1440) has been cited as Ref. 25 for the Vickers hardness value of bulk HfC (~18.3 GPa) in the revised manuscript.

Comment 4: If possible, please add the vendors of the carbide raw materials, tungsten cemented carbide media and pots.

Response: Thanks for the suggestion. The vendors of the carbide raw materials, tungsten cemented carbide media and pots have been added in the revised manuscript.

Modification: On page 13 (Methods part), the original sentence “The WC, TaC, NbC, ZrC and TiC powders with purity of 99.9% and particle sizes of 1-3 μm were used as the raw materials” has been revised to “The WC, TaC, NbC, ZrC and TiC powders (Shanghai Shuitian Material Technology Co., Ltd., China) with purity of 99.9% and particle sizes of 1-3 μm were used as the raw materials. In addition, the vendors of the tungsten cemented carbide media and pots have been added in the revised sentence “The monocarbide powders were mixed in equimolar fraction. Tungsten cemented carbide balls (Yiwu Hongzhou Trading Co., Ltd., China) with various sizes ($\phi 10$: $\phi 8$: $\phi 5$ mm = 1: 3: 6) were used as milling media in tungsten cemented carbide pots (Changsha Miqi Instrument Equipment Co., Ltd., China) filled with argon”.

Comment 5: Was the starting powders dry milled? During the planetary ball milling, was the rotation direction reversed after each cycle? How was the average grain size calculated based on the inverse pole figure? Please mention.

Response: Thanks for the comment. The starting carbide powders was dry-milled without adding any process control agents. During each cycle of the milling, the ball mill rotates forward for 25 minutes, pauses for 5 minutes, and then reverses for 25 minutes followed by a pause for 5 minutes.

As suggested by the reviewer, these important details have been added in the revised manuscript (Methods part).

The average grain sizes of the sintered HEC samples were determined based on EBSD measurements using TSL OIM analysis software. Electron backscatter diffraction (EBSD) is an SEM method that steps the beam across the sample surface by a defined amount (the step size) and determines the phase and crystallographic orientation of the sample at each step from the diffraction pattern produced when the sample is tilted at 70° to the horizontal. This information can be used to produce maps of the sample microstructure which can be evaluated using the crystallographic information to determine the relative location of grain boundaries and phases. In the current estimation, a circle equivalent diameter based on the grain area is regarded as the grain size, and the specific values are derived from the TSL OIM analysis software.

Modification: On page 13 (Methods part), the original sentence “The milling process was conducted using a planetary ball mill (MiQi, China) at 250 rpm for 110 h, with a pause of 5 min after every 25 min running to avoid combustion caused by overheating” has been revised to “The dry milling process was conducted using a planetary ball mill (Changsha Miqi Instrument Equipment Co., Ltd., China) at 250 rpm for 110 h. During each cycle of the milling, the ball mill rotates forward for 25 minutes, pauses for 5 minutes, and then reverses for 25 minutes followed by a pause for 5 minutes.” On page 14, the sentence “The average grain sizes of the sintered HEC samples were determined based on EBSD measurements using TSL OIM analysis software.” has been added.

Comment 6: For the HEC sintering, what was the heating and cooling rate? After SPS, how did the author remove the graphite foil? Please add.

Response: Thanks for the comment. During sintering, the sample was heated to 1800 °C with a heating rate of 85 °C min⁻¹, and held for 5 min, followed by cooling to room temperature at an average cooling rate of 466 °C min⁻¹ (as shown in supplementary Fig. 2). After SPS, the cylindrical samples with dimensions of ϕ 30 mm \times 8 mm were obtained. Graphite foils on the sintered sample surfaces were removed by grinding with diamond sandpapers.

Modification: On page 14 of the revised manuscript, the sentence “The heating and cooling rate were $85\text{ }^{\circ}\text{C min}^{-1}$ and $466\text{ }^{\circ}\text{C min}^{-1}$, respectively” has been added. The sentence “Graphite foils on the sintered sample surfaces were removed by grinding with diamond sandpapers” has also been added (on page 14).

Comment 7: How were the lattice parameter and the quantity of t/m-ZrO₂ calculated? Rietveld refinement based on XRD patterns? Please add in the Characterization section.

Response: Thanks for the suggestion. The lattice parameter and the quantity of t/m-ZrO₂ were calculated by the Rietveld refinement method based on XRD patterns. This related information has been added in the characterization section of the revised manuscript.

Modification: On page 14, the sentence “The Rietveld refinement method was used to determine the lattice parameters and the phase fractions using MDI Jade 6.0 software.” has been added.

Comment 8: Page 14 line 396, c is in Equation (2); line 404, K_{IC} should be calculated by Equation (5) and (6).

Response: We are very grateful to the reviewer for pointing out this. The two mistakes have been corrected in the revised manuscript.

Modification: In line 396 of page 14, the expression “c in equation (2)” has been revised to “c in equation (4)”. In line 404 of page 15, the original sentence “The K_{IC} was calculated by Equation (3) and Equation (4)” has been revised to “The K_{IC} was calculated by Equation (5) and Equation (6)”.

Comment 9: Revise the caption of Fig. 3. a-b is the SEM images of the what?

Response: Thanks for the comment. Figure 3 shows the microstructure and element distribution in the (WTaNbZrTi)C HEC sintered at 30 MPa. a and b are the high magnification SEM images showing the ZrO₂ particles in the HEC matrix.

Modification: On page 24, the original figure caption “Fig. 3. Microstructure of the (WTaNbZrTi)C HEC sintered at 30 MPa. a-b, SEM image of the” has been replaced by “Figure 3. Microstructure and element distribution in the (WTaNbZrTi)C HEC sintered at 30 MPa. a-b, High magnification SEM images showing the ZrO₂ particles in the HEC matrix”.

Comment 10: Page 6 line 149, (00-2)_{FCC}//(1-10)_{t-ZrO₂} was not given in Fig 4, remove it.

Response: We thank the reviewer for the suggestion. We agree and have removed the expression in the revised manuscript.

Modification: In line 149 of page 6, the expression “with an orientation relationship of (00 $\bar{2}$)_{FCC}//(1 $\bar{1}0$)_{t-ZrO₂} and [110]_{FCC}//[111]_{t-ZrO₂}” has been removed.

Comment 11: Page 12 line 322, please revise the sentence “The (WTaNbZrTi)C HEC ... 30 MPa”. The comma should be removed.

Response: Thanks for the suggestion. The comma in the sentence “The (WTaNbZrTi)C HEC sintered at 50 MPa with higher relative fraction of t-ZrO₂, has higher fracture toughness compared to that sintered at 30 MPa” has been removed.

Modification: On page 12, the original sentence “The (WTaNbZrTi)C HEC sintered at 50 MPa with higher relative fraction of t-ZrO₂, has higher fracture toughness compared to that sintered at 30 MPa” has been revised to “Accordingly, the (WTaNbZrTi)C HEC sintered at 50 MPa with higher relative fraction of t-ZrO₂ has higher fracture toughness compared to that sintered at 30 MPa”.

Reviewer #2

Interesting original manuscript which first time describing the concept of transformation toughening in bulk high-entropy ceramics by in - situ ZrO₂ grains! The work supports the conclusions, the methodology is OK with enough details with excellent HRTEM work

Response: We are grateful to the reviewer for the recognition of the novelty of our work and for the clear recommendation for the publication of our work. We have further improved the manuscript based on the editor's and reviewers' comments.

1. The title - Ultrahard bulk high-entropy carbides with enhanced toughness via metastable in-situ particles.....

According to me the hardness measured at 9.8 N with value 21 GPa is not ultrahard.... please see publications:

J. Dusza, T. Csanádi, D. Medved', R. Sedlák, M. Vojtko, M. Ivor, H. Ünsal, P. Tatarko, M. Tatarková, P. Šajgalík, Nanoindentation and tribology of a (Hf-Ta-Zr-Nb-Ti)C high-entropy carbide, J. Eur. Ceram. Soc. 41 (2021) 5417-5426A.

Naughton Duszová, L. Ďáková, T. Csanádi, A. Kovalčíková, V. Kombamuthu, H. Ünsal, P. Tatarko, M. Tatarková, P. Hvizdoš, P. Šajgalík, Nanohardness and indentation fracture resistance of dual-phase high-entropy ceramic, Ceram. Int., on line, <https://doi.org/10.1016/j.ceramint.2022.12.027>

reporting hardness of HE ceramics measured at 9.8 N with significant higher hardness.

Response: Thanks for the comment. We agree with the reviewer.

The term “ultrahard” is usually used for materials with hardness $H \geq 80$ GPa, comparable to that of diamond (80~100 GPa). In addition, one of the conventional criteria for defining a “superhard” material is that the hardness $H \geq 40$ GPa (e.g., nanocrystalline-BN ≈ 48 GPa). However, the hardness value is significantly dependent on the testing method and the applied load. The hardness testing methods for defining superhard or ultrahard materials have not been unified in literatures. For instances, the Knoop indentation hardness and Vickers hardness with applied force of 0.5 N have been used for bulk cubic-BN, ReB₂ and WB₄ materials in the work by Zhang, *et al.* (Adv. Mater., 2021, 33, e2005112), whereas nanoindentation hardness is commonly used for thin films. We also noticed that, in the excellent work by J. Dusza *et al.* (J. Eur. Ceram. Soc., 2021, 41: 5417-

5426A), the hardness of (HfTaZrNbTi)C HEC was measured by nanoindentation with a maximum penetration depth of 300 nm, instead of Vickers hardness. Accordingly, the nanohardness of 38.5 GPa can hardly be comparable to the Vickers hardness of 21.0 GPa for the (WTaNbZrTi)C HEC in this work.

Alternatively, many publications focus on materials with large elastic moduli (ultra-incompressibility) which are believed to be “intrinsically” superhard (S. Veprek, *et al.*, Chapter 6 Superhard and ultrahard nanostructured materials. 2016, 167-210, DOI 10.1007/978-3-319-29291-5_6). The criterion of ultra-incompressibility is that the bulk modulus $K > 300$ GPa (Michael T. Yeung, *et al.*, *Annu. Rev. Mater. Res.*, 2016, 46: 2.1-2.21). Tungsten carbide (WC) is one of the most incompressible materials, with a bulk modulus of $K = 421$ GPa. The (WTaNbZrTi)C HEC is also an incompressible material due to the high bulk modulus of $K = 281$ GPa based on density function theory (DFT) calculations (Ref. 26 in the revised manuscript). Therefore, the (WTaNbZrTi)C HEC can also be considered “intrinsically” superhard. We have revised the term “ultrahard” to “superhard” in the revised manuscript.

In some literatures the term “ultrahard” is even used for describing some steels or other alloys with hardness values much less than 10 GPa. Some of those publications are listed below:

Parisa Edalati, *et al.*, *Mater. Lett.*, 2021, 302: 130368.

Joshua A. Smeltzer, *et al.*, *Mater. Design*, 2021, 210: 110070.

However, as also suggested by the reviewer, the use of the term “ultrahard” in those cases is likely to be inappropriate.

Modifications: We have revised the title to: “Superhard bulk high-entropy carbides with enhanced toughness via metastable in-situ particles”. On page 1, we revised the associated sentences in the abstract: “Here, we introduce a strategy to achieve superhard HECs with enhanced toughness via in-situ metastable particles”, and “The work demonstrates the concept of using in-situ metastable particles for toughening bulk high-entropy ceramics by taking advantage of their compositional flexibility”. In addition, the above two publications listed by the reviewer have been cited in the revised manuscript as Ref. 10 and Ref. 11.

2. In the methods - For a comparative study, the fracture toughness was also determined using a universal testing machine (23 MTS Insight) based on the single edge notched beam (SENB, ASTM C 1421-18) method. The dimension of testing samples is 2 mm × 4 mm × 20 mm, with a pre-fabricated crack of 2 mm × 0.2 mm..... In this case it is not "crack" but notch, with a radius cca 0.1 mm, which is a little large value which can cause increased measured fracture toughness value.....

Response: We appreciate the reviewer for the concerns and we strongly agree to the reviewer's viewpoint. The effects of notch width and depth were examined for Al₂O₃, SiC and graphite materials by L.A. Simpson (J. Am. Ceram. Soc., 1974, 57: 151-154). The paper indicates that the wider notch results in higher fracture toughness value, but has varying degrees of impact on different materials. As shown in Simpson's work, at notch widths of 0.2 mm and 0.75 mm, the discrepancy of the fracture energy test results for Al₂O₃ is small, but significant for the SiC and graphite materials.

In an idea case, a beam containing a zero-volume crack should be considered. However, this geometric configuration is difficult to prepare in practice for the materials with high hardness and strength. At current technological level, it is difficult to prepare a crack/notch remarkably narrower than 0.2 mm. According to ASTM C 1421-10, advantages of such a "precracked" beam method (K_{Ipb}) are that it uses a classic fracture configuration and the "precracks" are large and not too difficult to measure, compared to the other two methods of K_{Isc} (surface crack in flexure) and K_{Ivb} (chevron-notched beam test).

According to a careful literature review, we found that the geometric configuration of our SENB samples has been accepted. For instance, in the work by Tan, *et al.* (Ceram. Inter., 2021, 47: 16882-16890), the fracture toughness (K_{IC}) of (HfTaNbTiZr)C and (HfTaNbVTi)C HECs were measured by the SENB method with sample dimensions of 2 mm × 4 mm × 36 mm, where the notch has a depth of 2 mm and a width of 0.2 mm, identical to that in our present work. The tested fracture toughness (K_{IC}) in that work is (HfTaNbVTi)C~4.8 MPa·m^{1/2} and (HfTaNbTiZr)C~4.4 MPa·m^{1/2}, remarkably lower than that of the (WTa NbZrTi)C HEC in our present work (5.89 MPa·m^{1/2}). In a work by Liu, *et al.* (Mater. Sci. Eng. A, 2021, 804: 140520), the fracture toughness of the (NbTaMoW)C HEC (K_{IC} ~3.6 MPa·m^{1/2}) was also tested by three-point bending with the sample dimensions of 2 mm × 4 mm × 20 mm, yet the paper did not show the notch size and the calculation procedures.

Overall, although it's hard to have SENB specimens with crack/notch width much narrower than 0.2 mm for the current superhard HECs, our current SENB testing results with the notch width of 0.2 mm are comparable to those in literature. Also, we have additionally provided the fracture toughness results measured from indentation method, which can also serve as references for comparison.

Modifications: On page 15, the original sentence: “The dimension of testing samples is 2 mm × 4 mm × 20 mm, with a pre-fabricated crack of 2 mm × 0.2 mm.” has been revised to “The dimensions of the SENB specimens are 2 mm × 4 mm × 20 mm (width × height × length), with a notch of 2 mm × 0.2 mm”. On page 15, the original sentence “*a* is the crack length” has been revised to “*a* is the notch depth”.

3. What is Your idea, how it is possible to optimize the ZrO₂ content with the aim to optimize the fracture toughness value?

Response: We appreciate the excellent comment raised by the reviewer. Our current work shows that the in-situ formed metastable ZrO₂ particles toughen the high-entropy carbides (HECs), and increasing the sintering pressure can promote this effect due to the higher fraction of tetragonal ZrO₂ phase. The work demonstrates the concept of using in-situ metastable particles for toughening bulk high-entropy ceramics by taking advantage of their compositional flexibility. In this regard, abundant future studies can be conducted by researchers in the community to further optimize the fracture toughness values of HECs, and several aspects can be considered as outlined in the following.

- 1) During the fabrication process of the HECs, increasing oxygen partial pressure in the sintering chamber is beneficial for the oxidation of Zr to form ZrO₂. However, it should be noted that excessive oxygen may lead to oxidation of other components and forming other types of detrimental oxides, e.g., WO₃, Ta₂O₅, Nb₂O₅ and TiO₂.
- 2) Refining the ZrO₂ particles in the high-entropy ceramic matrix is beneficial for achieving higher fraction of metastable tetragonal ZrO₂. As mentioned in our manuscript: “The reduction of particle size is also beneficial for the retention of t-ZrO₂ at room temperature” (page 11). In our work, the larger sized ZrO₂ particles at the HEC grain boundaries are more commonly to

be transformed into a monoclinic structure, whereas the smaller sized ZrO₂ particles (less than ~100 nm, and form a coherent interface with matrix) located in grain interiors usually remain the tetragonal structure after sintering. This can be realized by controlling the fabrication process, such as using properly lower sintering temperature, faster cooling rate and finer raw powders.

- 3) High-entropy ceramics with higher elastic moduli is more conducive to the retention of tetragonal ZrO₂ particles. It is generally acknowledged that martensitic transformation of t-ZrO₂ to m-ZrO₂ takes place when cooling from high temperature to ~1170 °C, accompanied by 3~5% volume expansion and 0.16 shear strain. Volume dilatation of the ZrO₂ particles upon cooling from the sintering temperature can be inhibited by the rigid high entropy ceramic matrix with high modulus, thus the higher content of tetragonal ZrO₂ can be obtained. For example, the ZrO₂ reinforced Al₂O₃ composites benefit from the fact that the elastic modulus of Al₂O₃ (390 GPa) is approximately twice that of ZrO₂ (207 GPa) (F.F. Lang, J. Mater. Sci. 1982, 17: 247-254).
- 4) For non-equimolar high-entropy ceramic materials, the relative fraction of Zr can be properly increased to optimize the ZrO₂ content. In the design of this type of transformation toughened high-entropy ceramics, it is worth noting that Zr should be the most prioritized component for oxidation. If components with higher activity such as Hf are included, the formed HfO₂ or (Hf, Zr)O_x may not have a significant toughening effect. For example, in the work by Wang, *et al.* (Acta Mater., 2022, 231: 117887), many (Hf, Zr)O₂ particles are formed in (Hf_{0.25}Ta_{0.25}Zr_{0.25}Nb_{0.25})C_{0.5}N_{0.5} sample.

Modification: A perspective for the future studies has been added in the revised manuscript (Summary part): “The work thus provides a useful strategy for toughening superhard HECs. For future efforts, the fracture toughness values can be further enhanced based on the present strategy by optimizing the design and processing of the HECs, e.g., adjusting the relative fraction of Zr in the non-equimolar variants, amending the oxygen partial pressure in the sintering chamber and refining the ZrO₂ particles in the HEC matrix”.

REVIEWERS' COMMENTS

Reviewer #1 (Remarks to the Author):

Dear Author,

Thanks for your revise. I am very satisfy about the current version. Great work.

Reviewer #2 (Remarks to the Author):

I am happy with the answers/corrections of the authors.

I recommend to publish the manuscript.